# Enhanced spin Seebeck effect signal due to spin-momentum locked topological surface states

Zilong Jiang[1], Cui-Zu Chang[2], Massoud Ramezani Masir[3], Chi Tang[1], Yadong Xu[1], Jagadeesh S. Moodera[2,4], Allan H. MacDonald[3] & Jing Shi[1]

Spin-momentum locking in protected surface states enables efficient electrical detection of magnon decay at a magnetic-insulator/topological-insulator heterojunction. Here we demonstrate this property using the spin Seebeck effect (SSE), that is, measuring the transverse thermoelectric response to a temperature gradient across a thin film of yttrium iron garnet, an insulating ferrimagnet, and forming a heterojunction with $(Bi_xSb_{1-x})_2Te_3$, a topological insulator. The non-equilibrium magnon population established at the interface can decay in part by interactions of magnons with electrons near the Fermi energy of the topological insulator. When this decay channel is made active by tuning $(Bi_xSb_{1-x})_2Te_3$ into a bulk insulator, a large electromotive force emerges in the direction perpendicular to the in-plane magnetization of yttrium iron garnet. The enhanced, tunable SSE which occurs when the Fermi level lies in the bulk gap offers unique advantages over the usual SSE in metals and therefore opens up exciting possibilities in spintronics.

[1] Department of Physics and Astronomy, University of California, 3401 Watkins Drive, Riverside, California 92521, USA. [2] Francis Bitter Magnet Lab, Massachusetts Institute of Technology, Cambridge, Massachusetts 02139, USA. [3] Department of Physics, University of Texas at Austin, Austin, Texas 78712, USA. [4] Department of Physics, Massachusetts Institute of Technology, Cambridge, Massachusetts 02139, USA. Correspondence and requests for materials should be addressed to J.S. (email: jing.shi@ucr.edu).

Topological insulators (TIs) are a newly identified class of band insulators that exhibit a variety of unusual phenomena associated with topologically protected metallic surface states[1,2]. The number of surface state bands in TI gaps is odd at any energy. Because Kramer's theorem implies that states with opposite surface momentum have opposite spin-orientation, this property entails that TI surface states exhibit strong spin-momentum coupling, especially so when the two-dimensional surface states have a single branch. The momentum-space spin texture of TI surface states has been confirmed by ARPES[3,4]. Recent experiments[5–7] have demonstrated that spin-momentum coupling in the band structure translates into exceptionally strong spin-galvanic effects[8] and that TIs are more efficient than any metal for magnetization switching by spin-orbit induced torques. Conversely, in spin pumping experiments[9–11], TIs have also been demonstrated to support strong inverse spin Hall effects.

While there is little doubt that TIs offers a clear advantage over conventional conductors in generating and detecting pure spin currents, it has been unclear whether the unusually large effects are dominated by bulk or surface TI states[6,7,9–11]. Discrepancies between experimental results from different groups, and between results from different samples under nominally identical conditions point to the importance of extrinsic conditions. Uncertainties related to the structure of the interface, including the possibilities of intermixing of magnetic elements[12] and proximity-induced ferromagnetism[13], can further complicate the interpretation of spin transport measurements. Therefore, to separate surface and bulk contributions, it is imperative to systematically tune the Fermi level relative to the bulk band gap[14].

Although pure spin current generation by heat is already established in bilayers consisting of a heavy metal and a ferromagnet, the detection of the spin current with topological surface states in TI has not been demonstrated. In this work, we address an unusual spin Seebeck effect (SSE), the efficient conversion of thermally driven magnons into an electromotive force (emf) due to TI surface states. To exclude from our measurements, the anomalous Nernst effect[15] induced emfs that result from the flow of spin-polarized charge in a ferromagnetic metal in SSE devices[16], we choose yttrium iron garnet (YIG), a ferrimagnetic insulator, as the source of magnons. In the longitudinal SSE configuration[17–19], a vertical temperature gradient in YIG drives a magnon current. In the steady state, magnon flow towards the TI/YIG interface is balanced by decay of excess magnons via either magnon–phonon scattering in YIG or magnon interactions with TI surface or bulk electrons. Our experiments demonstrate that the electrical consequences of the decay of magnon population excesses or deficiencies are particularly simple and strong for TI surface states.

## Results

**SSE from magnon-topological surface states interaction.** When a magnon is created or annihilated, an electron flips from majority to minority spin-orientation or *vice versa* to conserve total spin, as illustrated schematically in Fig. 1a. When the YIG magnetization is in the $\hat{y}$ direction, spin-momentum locking in the TI surface states then leads to a net rate of momentum transfer $\delta k_x$ in the $\mp \hat{x}$ direction, with the sign depending on whether the Fermi level lies in the surface state conduction or valence band. Because the relationship between momentum and velocity is opposite in the two bands, the opposite momentum transfer produces the same velocity direction of electrons. Hence, magnon relaxation via interaction with TI surface states leads to currents of the same sign for n or p surface states, or under open-circuit conditions to emfs of the same sign.

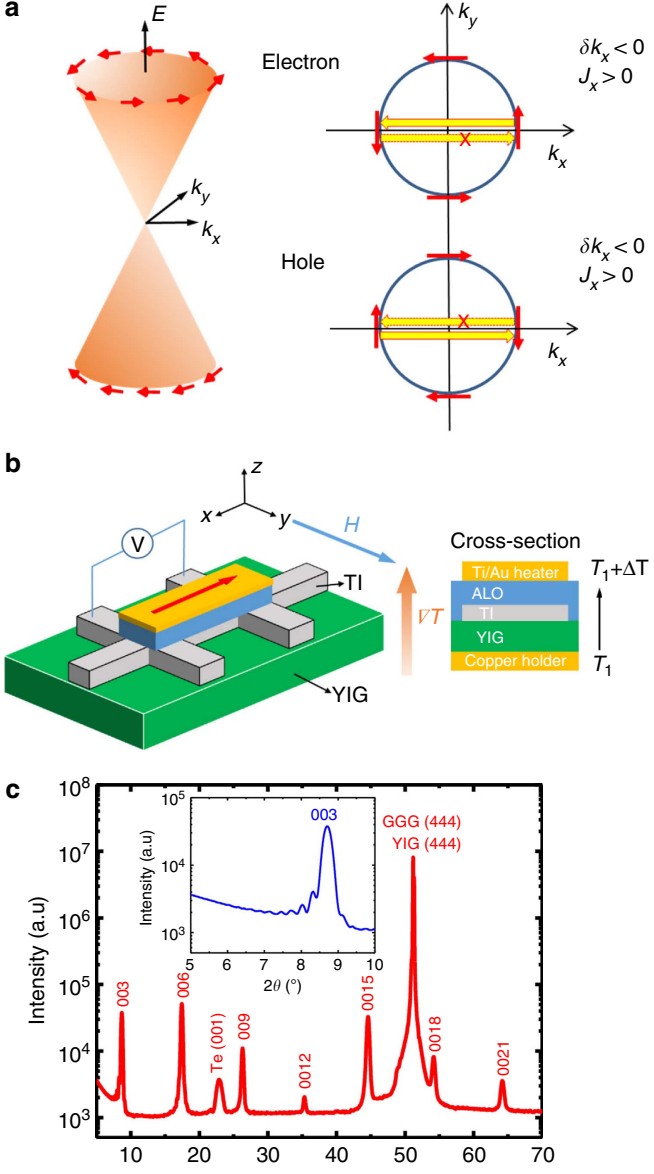

**Figure 1 | SSE from topological surface states and heterostructure sample properties.** (**a**) Dirac-model topological insulator Fermi surface. Assuming isotropic exchange interactions between YIG and the topological insulator surface states, electrons flip from majority to minority spin directions when a magnon is annihilated. For YIG magnetization in the y-direction, in the conduction band, magnon annihilation scatters electrons near the Fermi surface from $k_x$ to $-k_x$ directions but not from $-k_x$ to $k_x$, resulting in a net flow of electrons along $-k_x$ or a positive current $J_x$. The electron scattering amplitudes from y to $-y$ and from $-y$ to y associated with magnon annihilation are equal. In the valence band, magnon annihilation produces a positive $\delta k_x$, but a positive $J_x$ as well. (**b**) Device schematics for SSE experiments. The sample consists of a Hall bar structure TI film on YIG. An insulating $Al_2O_3$ layer covers the entire substrate. The current (red arrow) flows in the heater, producing a temperature gradient $\nabla T$ along the z-axis. The in-plane magnetic field is applied along the y-axis while the dc SSE voltage is measured along the x-axis. The side view shows the layered structure of the device. (**c**) X-ray diffraction result of a typical 20 QL-$(Bi_xSb_{1-x})_2Te_3$ grown on YIG/GGG. The inset shows a zoom-in view of the (003) peak and its associated Kiessig fringes.

We carry out longitudinal SSE experiments in TI/YIG heterostructures at room temperature. The TI is five-quintuple layer (QL)-thick $(Bi_xSb_{1-x})_2Te_3$ in which the Fermi level is tuned between bulk valence and conduction bands by changing Bi/Sb ratio[14] as indicated by the resistivity and the ordinary Hall data. The atomically flat YIG is grown first at $\sim 700\,°C$, and the TI is grown later at a much lower temperature ($\sim 250\,°C$) so that interface mixing is minimized. The $(Bi_xSb_{1-x})_2Te_3$/YIG heterostructure is therefore an excellent tunable system in which surface and bulk state contributions to the SSE can be disentangled.

**Heterojunction preparation and longitudinal SSE geometry.** YIG films that are 20-nm thick are grown on gadolinium gallium garnet (GGG) (111) substrates by pulsed laser deposition (PLD) as described previously[20–22]. Atomic force microscopy (AFM) measurements show that the surface roughness of YIG is $\sim 0.1\,nm$ over a scanned area of $2 \times 2\,\mu m$ (Supplementary Fig. 1). YIG/GGG has well-defined in-plane magnetic anisotropy. After high temperature annealing, a five QL $(Bi_xSb_{1-x})_2Te_3$ film is grown on top in an ultrahigh vacuum molecular beam epitaxy (MBE) system and capped by a 5-nm-thick epitaxial Te layer (Methods). A sharp and streaky reflection high-energy electron diffraction (RHEED) is present throughout growth, indicating a highly ordered TI film on YIG (111) with a smooth surface (Supplementary Fig. 3). Figure 1c shows an X-ray diffraction pattern for a 20 QL TI on (111)-oriented YIG/GGG grown under the same conditions, confirming its highly crystalline quality (the main figure appeared as Fig. 2b in ref. 23). Peaks can be identified with the (00n)

diffraction peaks of $(Bi_xSb_{1-x})_2Te_3$, with the YIG/GGG (444) peak or with the (001) peak of the Te capping layer, suggesting that the film is grown along the c axis and that no impurity phase is present. The zoom-in view of the (003)-peak (Fig. 1c inset) shows multiple Kiessig fringes on both sides, demonstrating the excellent layered structure of the TI films and good TI/YIG interface correlation. More structural characterization data can be found in ref. 23.

As schematically shown in Fig. 1b, the heterostructure sample is patterned into a $100 \times 900\,\mu m$ Hall bar structure with Ti(5 nm)/Au(80 nm) contacts (Methods). A 150 nm $Al_2O_3$ insulating layer is subsequently deposited by atomic layer deposition (ALD). A 100-μm-wide Ti(5 nm)/Au(45 nm) strip vertically aligned with the Hall bar channel is defined on top of the $Al_2O_3$ layer to form a heater. In the longitudinal SSE experiment, we turn on the heater with a dc current of varying magnitude to generate a vertical temperature gradient $\nabla T$ ($//\hat{z}$). The TI surface states provide a decay channel for non-equilibrium YIG magnon populations, which generate a dc voltage $V_{SSE}$ in the TI layer when active. A closed-cycle refrigerator is used to keep the sample temperature constant when the heater is on. As the in-plane magnetic field ($//\hat{y}$) is swept perpendicular to the main Hall bar channel ($//\hat{x}$), a $V_{SSE}$ hysteresis loop is recorded.

**Elimination of anomalous Nernst contribution.** In structures containing a magnetic-insulator and a spin-orbit coupled conductor, the longitudinal SSE can be mixed with the anomalous Nernst effect from the induced magnetic layer[18,19]. To judge whether the voltage we measure should be interpreted as a SSE or

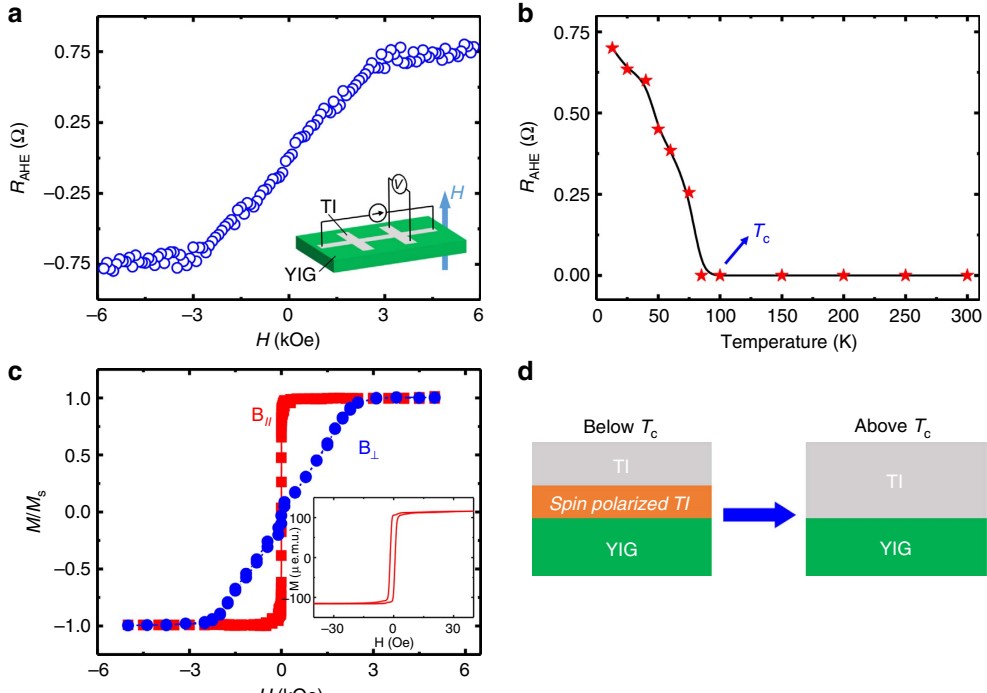

**Figure 2 | Proximity-induced magnetization in TI/YIG at low temperatures. (a)** A typical anomalous Hall curve for a five QL $(Bi_xSb_{1-x})_2Te_3$/YIG sample ($x = 0.24$) at 13 K. The inset shows schematic illustration of Hall measurements. **(b)** Temperature dependence of the anomalous Hall resistance for the same sample, indicating that the mean surface-state proximity-induced exchange splitting drops below $k_BT$ at $\sim 100$ K. This temperature may be viewed as an effective critical temperature for proximity-induced surface-state magnetization. **(c)** 300 K VSM magnetic hysteresis loops for both in-plane and perpendicular magnetic fields. The inset shows a low-field in-plane hysteresis loop with a coercive field $\sim 2$ Oe. **(d)** Schematic illustration of the SSE signal free from the proximity-induced anomalous Nernst effect. Below $T_c$, there is a spin-polarized TI interface layer, which may produce an anomalous Nernst signal mixed in the SSE signal. The spin-polarized TI layer does not exist above $T_c$.

anomalous Nernst voltage from charge current flow through a partially magnetized TI, we first address the strength of proximity-induced ferromagnetism in the TI. Figure 2a shows the nonlinear contribution to the total Hall data in a five QL $(Bi_xSb_{1-x})_2Te_3$/YIG sample ($x = 0.24$) at $T = 13$ K, after removing the dominant linear ordinary Hall background. Vibrating sample magnetometry (VSM) data from a representative YIG film measured at 300 K are displayed in Fig. 2c. The shape of the nonlinear Hall signal in $(Bi_xSb_{1-x})_2Te_3$/YIG heterostructure resembles that of the YIG out-of-plane hysteresis loop except that the low-temperature saturation field is slightly higher. As discussed in ref. 24, a nonlinear Hall signal indicates an anomalous Hall effect (AHE)[25] arising from a magnetized TI surface layer, suggesting that conducting states at the TI/YIG interface participate strongly in the magnetic order. Additional data and evidence can be found in Supplementary Figs 4 and 5 and in ref. 26. The magnitude of the AHE as a function of temperature is presented in Fig. 2b. The AHE signal is unobservable above $\sim 100$ K. Other TI/YIG samples also show AHEs of varying strength that are observable up to $\sim 150$ K. We conclude that at 300 K, where we perform the longitudinal SSE experiments, the mean exchange energy experienced by the TI surface states is negligibly small compared with $k_BT$ ($T = 300$ K), and that the voltage we measure is free of anomalous Nernst contamination in all TI/YIG samples (Fig. 2d).

**Heterojunction SSE results.** Figure 3a shows a typical $V_{SSE}$ voltage measured as a function of applied magnetic field in the $\hat{y}$ direction ($\theta = 0$) for a $x = 0.24$ sample. The $V_{SSE}$ signal exhibits a hysteresis loop that is consistent with the low-field in-plane VSM loop (Fig. 2c inset). As the magnetization is reversed by the $y$-axis

field, the sign of $V_{SSE}$ is also reversed. On the other hand, the $V_{SSE}$ signal is absent when $H$ is swept along the $\hat{x}$ direction (Fig. 3a, $\theta = 90°$), consistent with Fig. 1a.

Although a precise determination of the temperature difference $\Delta T$ across the YIG film is difficult, it should be directly proportional to the heater power, $\Delta T \propto P = I^2R_{heater}$, where $I$ and $R_{heater}$ are, respectively, the current and resistance of the aligned Ti/Au heater. Figure 3b displays the $V_{SSE}$ hysteresis loops as a function of currents in the $x = 0.24$ sample. As the current increases, the $V_{SSE}$ hysteresis loop progressively increases in magnitude. Figure 3c shows $V_{SSE}$ as a function of the heater power. Clearly, $V_{SSE}$ is proportional to the heater power as expected for a thermally driven magnon transport phenomenon.

Both surface and bulk electrons in TI experience strong spin-orbit coupling; therefore both can contribute to $V_{SSE}$ in TI/YIG. To probe their relative contributions, we have investigated five $(Bi_xSb_{1-x})_2Te_3$/YIG heterostructure samples with different Bi fractions, $x = 0$, 0.23, 0.24, 0.36 and 1. As $x$ is varied, the Fermi level position is systematically tuned, as is the relative weight of the surface and bulk magnon relaxation processes. Figure 4a displays the temperature dependence of the longitudinal resistance, $R_{xx}$, for these five samples. The resistance data indicate metallic behaviour in both $Sb_2Te_3$ and $Bi_2Te_3$ (that is, for $x = 0$ and $x = 1$) and a smaller $R_{xx}$ than in the other three samples. In the $x = 0.36$ sample, $R_{xx}$ increases and has insulator-like temperature dependence. In the $x = 0.23$ and 0.24 samples, the resistance behaviour is more strongly insulator-like. For these samples, $R_{xx}$ increases with decreasing temperature over the entire temperature range, reflecting the depletion of bulk carriers. The five samples undergo a metal–insulator–metal crossover as $x$ increases from 0 to 1. Figure 4a inset depicts qualitatively the Fermi level position relative to the Dirac point at different Bi fractions inferred from ordinary Hall effect measurements (Fig. 4d).

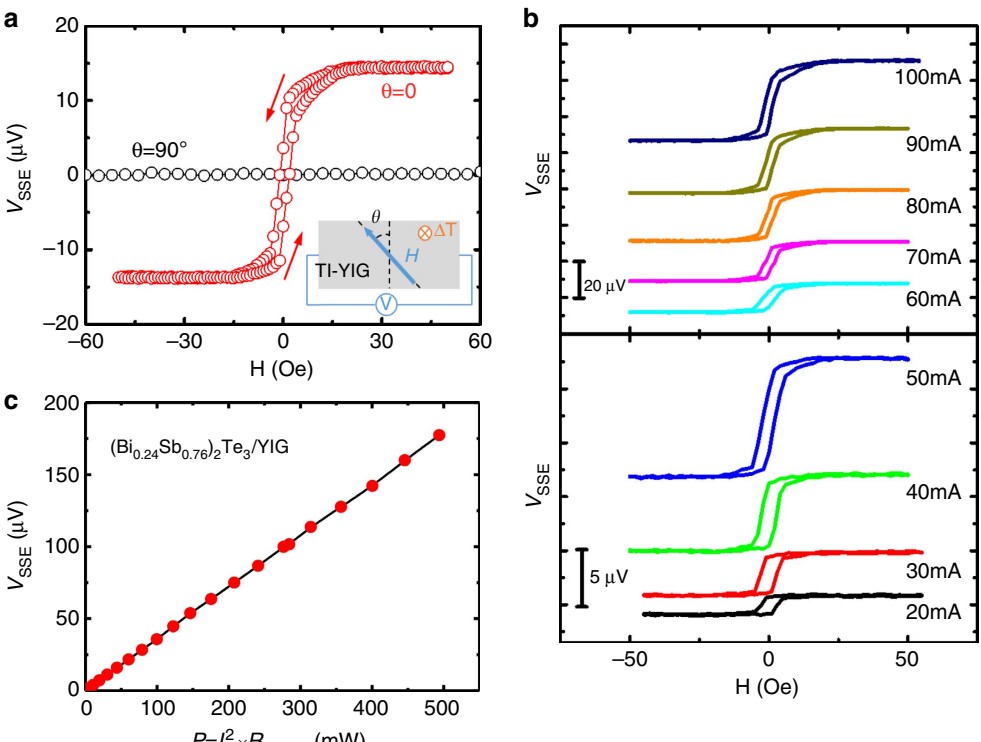

**Figure 3 | Observation of SSE in TI/YIG. (a)** A typical $V_{SSE}$ hysteresis loop in a 5 QL $(Bi_xSb_{1-x})_2Te_3$/YIG sample ($x = 0.24$) at room temperature. The heater current is 80 mA and the magnetic field is applied along two different directions ($\theta = 0$ or 90°). **(b)** $V_{SSE}$ loops at different heater powers by adjusting the heater current. **(c)** Heater power dependence of $V_{SSE}$ in TI/YIG sample at $H = 50$ Oe.

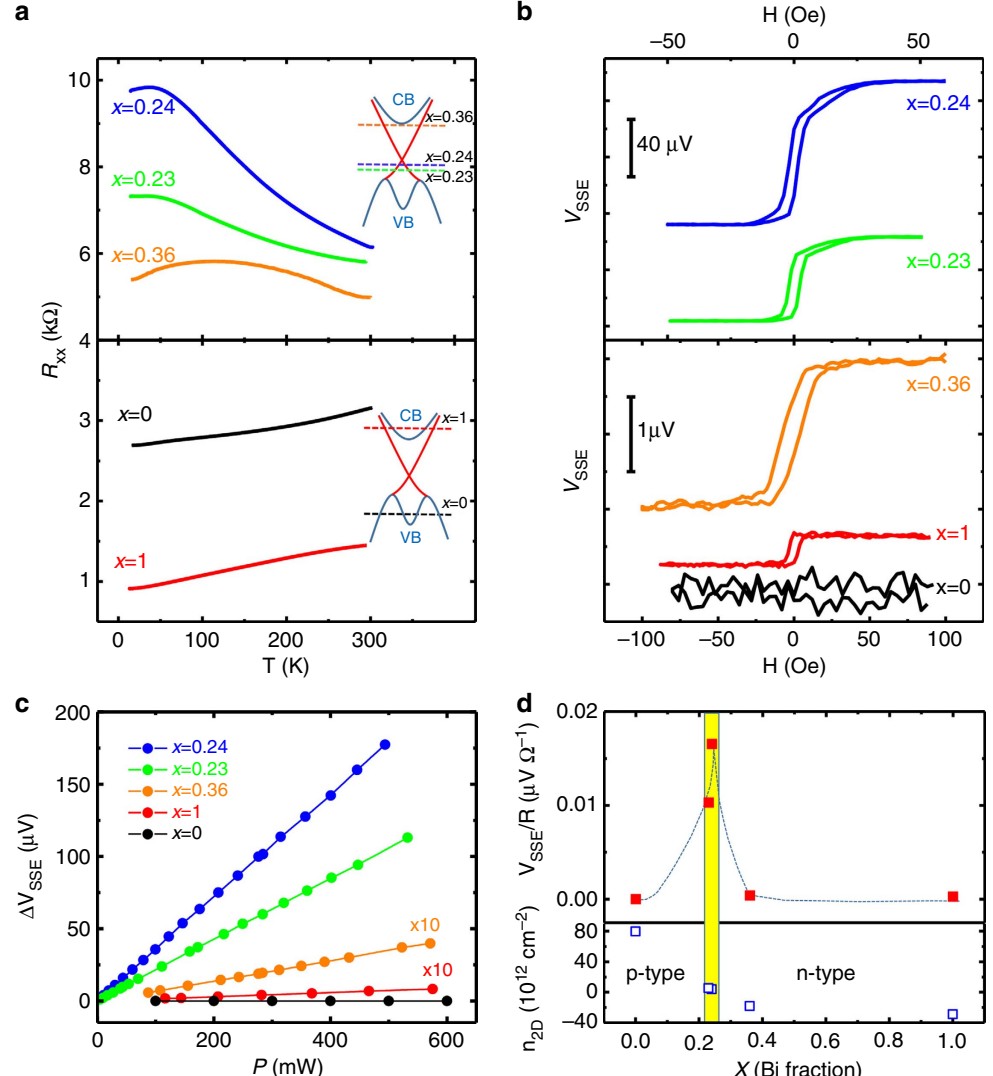

**Figure 4 | Giant SSE in TI when the Fermi level is tuned to the bulk gap.** (**a**) Temperature dependence of longitudinal resistance $R_{xx}$ for five QL $(Bi_xSb_{1-x})_2Te_3$/YIG samples with $x$ varying from 0 to 1. The inset shows schematic electronic band structures of $(Bi_xSb_{1-x})_2Te_3$ with the Fermi level at different values of $x$. The Fermi level position with respect to the Dirac point is tuned as the doping level is varied. (**b**) Field dependence of $V_{SSE}$ in five QL $(Bi_xSb_{1-x})_2Te_3$/YIG samples for different $x$ values under a fixed heater power. (**c**) Heater power dependence of $V_{SSE}$ in five QL $(Bi_xSb_{1-x})_2Te_3$/YIG samples with various $x$ values. (**d**) $V_{SSE}/R_{xx}$ and two-dimensional carrier density with various $x$ values.

**Enhanced SSE signals from surface states**. The five devices were all fabricated using nominally identical processes and have identical dimensions; therefore, their $V_{SSE}$ data can be quantitatively compared. Figure 4b reveals a striking contrast in $V_{SSE}$ values measured at a fixed heater power ($P = 283$ mW) between the metallic and insulating samples. $V_{SSE}$ is only 0.41 µV for $Bi_2Te_3$ ($x = 1$), increases to 1.94 µV for $x = 0.36$, and then rises precipitously to 100 µV for $x = 0.24$. The Fermi level at this point has just passed the Dirac point and the carrier type has switched from electrons to holes with a relatively low carrier density, $n_{2D} = 4 \times 10^{12}$ cm$^{-2}$. Note that the magnitude of $V_{SSE}$ at $x = 0.24$ is ~200 times greater than for the $Bi_2Te_3$ sample ($x = 1$), in which the electronic density-of-states is dominated by bulk conduction band states. For $x = 0.23$, $V_{SSE}$ is then reduced to 60 µV. For $x = 0$, or $Sb_2Te_3$, the $V_{SSE}$ signal is too small to be resolved, and its Fermi level intersects the bulk valence band (Fig. 4a inset) with the measured hole density of $8 \times 10^{13}$ cm$^{-2}$. The dramatic disparity between metallic and insulating samples reveals the overwhelming importance of the topological surface states in generating a SSE. In comparison, the SSE signal from a topological surface dominated TI sample is about 1 order of magnitude greater than that from a Pt/YIG device (Supplementary Fig. 6).

Figure 4c displays the $V_{SSE}$ voltage versus heater power for all five samples with different values of $x$, demonstrating that a linear relation holds for all samples. In Fig. 4d, we plot $V_{SSE}/R_{xx}$ versus $x$, demonstrating that the charge current induced by magnon decay is also greatly enhanced when surface states dominate. For example, $V_{SSE}/R_{xx}$ is enhanced by a factor of 50 when the Bi doping varies from $x = 1$ to 0.24. Such a large change does not seem to be caused by random sample-to-sample variations since all samples are grown and prepared by the same procedures. As shown in Supplementary Fig. 7, even for Pt/YIG samples prepared at different times, the $V_{SSE}/R_{xx}$ ratio only varies by ~5%. The stark contrast between the surface and bulk electronic states of TI must stem from the differences in magnon–electron relaxation. Although bulk TI states are strongly spin-orbit coupled, electronic majority to minority spin-flip processes do not always have the same sign of momentum transfer, which consequently suppresses the SSE emf.

## Discussion

We attribute the pronounced SSE at Fermi levels in the gap to competition between magnon population relaxation due to bulk electronic transitions, which do not yield a significant emf and magnon relaxation by surface electronic transitions which do yield a large emf. In the steady state, the magnon current towards the interface, $I_{MAG}$, is related to the excess magnon density at the interface, $N_{MAG}$, by $I_{MAG} = N_{MAG} (\tau_0^{-1} + \tau_S^{-1} + \tau_B^{-1})$ where $\tau_0^{-1}$ is the surface magnon relaxation rate in an isolated YIG film, $\tau_S^{-1}$ is the rate due to interactions with TI surface states and $\tau_B^{-1}$ is the rate due to interaction with TI bulk states. Assuming that only the surface state interaction leads to a substantial SSE voltage, we conclude that $V_{SSE}$ is proportional to $I_{MAG} \tau_S^{-1} / (\tau_0^{-1} + \tau_S^{-1} + \tau_B^{-1})$. The reduction in $V_{SSE}$ when the bulk relaxation mechanism is activated suggests that when present it is stronger than $\tau_0^{-1}$. The fact that the surface mechanism leads to a large effect suggests that it can also dominate over non-equilibrium magnon population decay mechanisms that are intrinsic to YIG films, even when the Fermi level lies close to the Dirac point and the surface density-of-states is relatively small.

In summary, we have observed a giant SSE voltage in topological surface states as the Fermi level in the TI is tuned to the bulk band gap. We explain this phenomenon in terms of spin-momentum locking in TI surface states that serve as a highly effective channel of magnon population decay at heterojunctions between magnetic and TI. This special SSE in TI/YIG heterostructures does not only yield a much larger SSE voltage than in structures consisting of heavy metals, but also offers unique tunability that the metallic systems do not have.

## Methods

**Heterojunction growth and characterization.** Thin YIG films are grown on polished single crystal GGG (111) substrates via PLD. The base pressure of the deposition chamber is $\sim 6 \times 10^{-7}$ Torr. During the growth, the substrate is heated up to 700 °C and the chamber is back filled with ozone ($\sim 1.5$ mTorr). The layer-by-layer growth mode and film thickness can be monitored and recorded by the RHEED pattern and its intensity oscillations (Supplementary Fig. 1a inset). YIG shows an epitaxial relation with the GGG substrate (Supplementary Fig. 3a). AFM image of a typical $\sim 20$ nm YIG film (Supplementary Fig. 1) indicates a root-mean-square (r.m.s.) roughness $\sim 0.1$ nm over a $2 \times 2$ μm scanned region. To characterize the magnetic properties of YIG films, the ferromagnetic resonance (FMR) data are taken with a frequency of 9.6 GHz as shown in Supplementary Fig. 2. The red solid line shows a fit to a Lorentzian function, with the FMR resonance field $H_{res} \sim 2{,}473$ Oe and line width $\Delta H \sim 6.5$ Oe. $4\pi M_s \sim 2{,}200$ Oe can be calculated from the Kittel equation. These results suggest that the YIG films are of high quality. To form heterostructures, YIG (111) films are then transferred to a custom-built ultra-high vacuum ($< 5 \times 10^{-10}$ Torr) MBE system for TI growth. To ensure good interface quality, *in situ* high temperature annealing (600 °C, 30 min) is performed to degas before film growth. The RHEED pattern is taken again to ensure same excellent quality of YIG surface (Supplementary Fig. 3b) after annealing. High-purity Bi (99.999%), Sb (99.9999%) and Te (99.9999%) are evaporated from Knudsen effusion cells. During the growth, the YIG substrate is kept at 230 °C and the growth rate is $\sim 0.2$ QL per min. The epitaxial growth is monitored by the *in situ* RHEED pattern. The sharp and streaky diffraction spots indicate a very flat surface and high-quality crystalline TI thin film grown on YIG (111) (Supplementary Fig. 3c). The film is covered with a 5-nm Te protection layer before taken out of the MBE chamber.

To fabricate SSE devices, the heterostructure is patterned into the Hall bar geometry by standard optical photolithography and Ar-plasma etching. A 5-nm Ti/80 nm Au is deposited by e-beam evaporation as Hall bar contacts. The sample is later loaded into an ALD chamber for $Al_2O_3$ insulating layer growth. Finally, a vertically aligned heater (Ti/Au) on top of $Al_2O_3$ is defined to form the SSE device. The experiment is taken in a close-cycle refrigerator equipped with an electromagnet (field up to 0.8 T).

**SSE measurements.** SSE is usually studied in two different configurations: transverse[16] and longitudinal[17–19]. Several methods can be used to generate a temperature gradient across the heterostructure, that is, by sandwiching the sample with two copper blocks as heat source and sink, respectively[17], by local laser heating or by applying a current through a normal metal (Joule heating). With Joule heating, it is possible to generate a controllable and uniform temperature gradient over the entire sample area quite effectively. In this work, we carry out the

longitudinal SSE experiments with an improved current heating method. In our case, a Ti/Au strip is defined on top of the Hall bar device, which is electrically insulated by the $Al_2O_3$ layer from the TI sample beneath, serving as an external heater. By sending a current through the heater, the device establishes a vertical $\Delta T$, which is adjustable by changing the current. To determine the exact sample temperature, we measure the sheet resistance of the TI film along the $\hat{x}$ direction of the Hall bar and compare with the calibrated $R_{xx}$ versus temperature curve. In all SSE measurements carried out with different heating powers, the sample temperature was always fixed at 300 K using a close-cycle refrigerator.

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

## Acknowledgements

We would like to thank V. Aji, H.W. Liu, X.C. Xie, Q. Niu, Y. Fan, J. Li and P. Wei for fruitful discussions, and W. Beyermann, D. Humphrey, M. Aldosary, B. Yang and R. Zheng for their technical assistance. YIG growth, device fabrication, measurements of magnetic and transport properties, SSE experiments and theoretical analyses at UCR and UT-Austin were supported as part of the SHINES, an Energy Frontier Research Center funded by the US Department of Energy, Office of Science, Basic Energy Sciences under Award #SC0012670. TI growth and X-ray diffraction were performed at MIT and

supported by the STC Center for Integrated Quantum Materials under NSF Grant No. DMR-1231319, NSF DMR Grants No. 1207469 and ONR Grant No. N00014-13-1-0301.

## Author contributions

J.S. designed and supervised the project. Z.J., Y.X. and J.S. designed the experiments. C.T. and Z.J. performed the pulsed laser deposition of YIG films and characterization. C.-Z.C. and J.S.M. grew the TI films with molecular beam epitaxy and performed x-ray diffraction measurements. Z.J. fabricated the SSE devices and performed the transport measurements and data analysis. M.R.M. and A.H.M. performed the theoretical analyses and modelling. Z.J., J.S. and A.H.M. wrote the manuscript and all the authors contributed to the final version of manuscript.

## Additional information

**Competing financial interests:** The authors declare no competing financial interests.

