## [Peer review file · Nature Communications]

All reviewers recommend publication of our paper, with the exception of referee #4. The concerns of referee #4 are mainly about itself. In this response we have explained our motivation for the terminology we use, and revised the MS to help ensure that readers understand the terminology. In fact the referee could as easily have directed his terminology complaints against the entire SSE field and we would not disagree too strongly. However the SSE terminology is now well established and the Spin Seebeck effect is destined to stand among the many examples of scientific effects which are somewhat inappropriately named. We trust that our paper can now finally be published. Detailed responses to the referee comments follow.

Reply to reviewers' comments (original in blue and reply in black)

Reviewer #1 (Remarks to the Author):

I still find the paper quite interesting and think that the authors have answered most questions reasonably. I would have liked to see the angular dependence of SSE just as a sanity check, I guess any reasonable experimentalist would measure more than two points. I am also a bit annoyed by the fact that the authors did not cite their own paper before which in fact contains a lot of at least similar physics as referee two has argued. Nevertheless I find the paper reasonably interesting and I guess it could be published in Nature Materials. I would be very interested in particular in the response of referee three.

Reply to Reviewer #1:

Thanks for the recommendation. The response of reviewer #3 shows that he/she has no more questions.

About the angular dependence, I understand reviewer's concern. As I explained in the last reply, this could have been done with a magnet that can rotate. If the SSE signal in two orthogonal directions had been non-zero, then we would have definitely done the rotation experiments to make sure the magnetic field is aligned with the width direction of the Hall bar (which we called the "y" direction). In the very first paper by K. Uchida et al. (Fig. 4a of Nature 455, 778 (2008)), they did show the angular dependence with measurements at 8 angles, in order to justify the SSE origin of their observations. In the second major paper by the same authors (Fig. 2g of Nature Materials 9, 894 (2010)); however, they only showed two hysteresis loops in two orthogonal directions, as in our manuscript. It is understandable that the vector product vanishes if the magnetic field is set along the SSE signal detection direction. Similarly, in our

previous SSE paper (Xu et al. Appl. Phys. Letts. 105, 242404 (2014)), we carried out full angular dependence measurements (Fig. 1b). Because we viewed the symmetry of the effect we were observing as well-established, we did not spend too much time verifying that the field is set precisely in the direction perpendicular to the direction in which the SSE voltage is measured in. In Fig. 3a of the present manuscript, the SSE voltage is zero at $\theta=90$. Therefore, we did not perform a tedious rotational experiment to verify the expected sine-function angle-dependence.

In the last revision, our recently published Nano Letters paper was added and cited as ref. 25. As one can see, the emphasis of ref. 25 is on induced ferromagnetism in a set of heterostructures. This present work, on the other hand, is intentionally carried out in the regime where induced ferromagnetism does NOT occur; therefore, the physics here is entirely different. When we first submitted this manuscript to Nature Materials in July, ref. 25 was still under review of Nano Letters and the fate of this submission was not known. Of course, it is in our best interest to cite our own work and we should have probably done so in the first version.

Reviewer #2 (Remarks to the Author):

In the reply, the authors explain their model based on the experimental facts and clarify the reason why only they could successfully observed the spin Seebeck effect (SSE). I agree that the temperature dependence of the resistance is a possible indicator to distinguish the surface transport from the bulk one. In their data, the SSE signal is enhanced only when the samples show the insulating behavior. I understand that they carefully prepared the samples, which are different from the sputtering NiFe on a topological insulator (TI). This point should be stated more strongly in the main text.

[1] Regarding the anomalous Hall effect (AHE) and/or the anomalous Nernst effect (ANE), however, I do not understand their logic. The authors define the Curie temperature (T_c) by disappearance of the AHE. (Although this definition is misleading, I will adopt it below.) The T_c is obtained under the magnetic field perpendicular to the film. However, the perpendicular magnetic-anisotropy is three order larger than the parallel ones. Hence, the T_c in the magnetic field parallel to the film must be much larger than the present value, $\sim 100K$. If this was correct, the AHE/ANE could survive in higher temperatures. It is not fair to say that there is no AHE/ANE above T_c using the data in Fig. 2 (a). Data about the AHE or a magnetic proximity in a magnetic field parallel to the film is required.

[2] In addition to this, the authors say, "Heat flows along the z-direction in the SSE measurements here, but the z-component of the momentum in 2D electron systems is quantized." I am confused by this. Where is your surface state? In my knowledge, the present material is the three dimensional TI. If the size effect became prominent due to the thin film, the surface state must be also changed. This point needs to be clarified.

[3] Regarding previous studies, the authors say, "... in those works, they simply used metallic TI...". This is not fair. Previous studies also use the insulating TI, which is checked by the resistivity (good indicator).

I need clarification about these points to judge recommendation for publication.

Reply to Review #2:

About the indicator of the surface states in transport, it is understood in the TI community that insulating-like resistance behavior (i.e. negative temperature coefficient) with low-temperature saturation can serve as a good indicator. More elaborate experiments such as the Shubnikov-de Haas oscillations can be done to achieve more quantitative characterization, but these require high magnetic fields and rotation of the field.

About the first point, AHE requires a perpendicular magnetic field, just as the ordinary Hall effect does. It is true that an in-plane field saturates the magnetization much more easily as the reviewer pointed out. However, the saturated in-plane magnetization would only produce the so-called "planar Hall effect" along some directions which is often difficult to interpret (depending on angles). On the other hand, spontaneous magnetization should still exist at zero applied magnetic field, and vanishes at and above T_c . Therefore, it is not the total magnetization (here the AHE signal which is proportional to the magnetization), but the remanent magnetization that should be used to establish spontaneous ferromagnetic order. Practically, it must be obtained by extrapolating the high field magnetization (or AHE here) to zero applied field, which is precisely how it was done here (the linear background has already been removed).

About the second question, the reviewer seems to agree that when the samples show an insulating behavior, the bulk conduction is suppressed; therefore, the surface states dominate. In those samples, we can treat the samples as bulk insulators with two-dimensional metallic surface states (probably about 1nm thick). I do not understand reviewer's statement: "If the size effect became prominent due to the thin film, the surface state must be also changed." The surface states themselves are two-dimensional and they are not affected by the thickness of the sample (bulk states are) or the lateral dimensions of our devices (many microns in length and width). Although we do not fully understand what point the reviewer wishes to make here, we have carefully edited our paper in an effort to make certain that our terminology is unambiguous.

The third comment is a fair one. Among several cited previous works, ref. 11 by Shiomi et al. did include some insulating samples along with metallic ones. Thanks for pointing this out.

Although I mentioned this in my reply to the first round of comments, nowhere in the manuscript indicates that those samples were all metallic. In the manuscript, we simply wished to emphasize that none of these referenced works was about the spin Seebeck effect.

I want to thank the reviewer for his/her time and effort in providing the comments and questions.

Reviewer #3 (Remarks to the Author):

The authors have given an extended reply to the issues brought up in the previous referee round. In response to the remarks made, they have corrected some errors, clarified some issues and also revised their manuscript, and in particular softened the claims made in it, as well as changed the title.

As far as I can judge most, but maybe not all, issues have been resolved. However, this is a very active area, which brings together new hybrid materials systems with new physics concepts, and it is not so easy to have all of this fully under control and understood at once. I can therefore now give a (modest) recommendation for publication.

Reply to Review #3:

Thanks for the comments and the recommendation!

Reply to Reviewer #4

Note: This review was lengthy so we have responded separately to different points raised by the referee. The referee's remarks mainly concern terminology. We have made revision in our manuscript to ensure readers understand the meaning of our terminology.

Reviewer #4 (Remarks to the Author):

I had a look at the revised manuscript and the referees have improved some of the detailed points, which I commend.

From the rebuttal overall I get the impression that the paper really not about the SSE, but about the spin current to charge signal conversion mechanism as explained by the authors:

"As I argued above, we were interested in disentangling contributions from different sources, bulk and surface states, which has never been done by any group."

So the SSE is only used to generate a spin current and the study could also be done with other sources of spin current and maybe then the title is not very appropriate. I would then make this also clear in the manuscript.

We agree with the reviewer that our paper is really about the observation and interpretation of spin-current to charge signal conversion mechanisms at interfaces between magnetic insulators and topological insulators. Terminology is difficult however – and there is always a lot of room for differences of opinion. Obviously we are more interested in having our work published than in insisting on the terminology we employ in our title and elsewhere – but we do wish to explain why we have chosen this terminology. In our opinion the terminology ‘spin Seebeck effect’ is misleading. The name SSE suggests that the observed voltages are due to a difference between thermoelectric transport coefficients of majority and minority spins in a magnetic conductor, as proposed in the original experimental work by Uchida et al. The community now understands that the effect is in fact much richer and uses the term (as we understand it) to refer to emf’s generated via any mechanism which involves the influence of temperature gradients on degrees of freedom that carry spin, usually magnons or electrons. Our terminology ‘topological SSE’ is intended to briefly convey to readers that this is one of those types of emfs AND that it involves in an essential way the peculiar properties of topological insulators. Indeed we also considered as a title: ‘Spin-Charge Conversion at Topological Insulator Heterojunctions.’ We still believe that our present terminology conveys more of the effect we have observed, but we are open to suggestions.

I have a few other comments to the rebuttal. The authors claim: "But the SSE had never been studied in TI." However there is no indication that a SSE is present inside the TI! The SSE is the conversion of a temperature gradient and resulting heat current into a spin current and this occurs inside the ferromagnetic YIG. There is still a misconception with the authors, as the SSE has nothing to do with the ISHE or other spin to charge conversion mechanisms.

This is a continuation of the discussion of terminology above. The Spin Seebeck effect, as the people who first reported the effect called it, is NOT about generating a spin current in YIG or any other ferromagnets. It is obvious that a temperature gradient in a ferromagnet generates a spin current because magnons carry spins. It is not necessary to ‘convert an energy current into a spin current’. Our work here is about using efficient magnon/surface Dirac electron relaxation, as indicated in the title and the manuscript. The reviewer probably knows about the development of the research topic. SHE/ISHE is particularly interesting and relevant to SSE, spin pumping, SMR, spin-orbit torque, etc. These phenomena only exist in the detecting materials with strong spin-orbit coupling. In YIG, generating a spin current is given, but the relaxation mechanism is not known. In 3D detector materials, people can explain the SSE by ISHE, but such an explanation is not applicable in TIs. What we found in this work is that the surface states/magnon relaxation is much more efficient than the 3D-like bulk states. We have established the voltage generation mechanism by controlling the Fermi level position in the TIs.

The spin current and resulting spin accumulation due to the SSE in the ferromagnet (YIG) can also be detected for instance by optical techniques including BLS or other techniques not related to any adjacent layer. The authors should make it clear that they are aware what the SSE is and it clearly is distinct from the spin current to charge signal conversion mechanism and there is clearly nothing topological about the SSE in the YIG.

Again this point is about terminology. Our views are discussed above. Magnon detection by BLS or other means would be fun to do but is irrelevant to what we report here. Spin current generation by heat is an outcome of thermodynamics inside YIG if the relaxation mechanism is known. We agree that there is nothing topological about it.

So I am glad to see that the authors have revised the title (which however should may be changed further to focus more on the detection and not the spin current generation by the SSE). I agree that the spin current at the surface of the YIG vanishes if there is no adjacent layer, however then there is a spin accumulation that clearly results from the SSE. One does NOT need to consider the detector material because one can detect the spin current and resulting spin accumulation without a detector layer by other means (optical, etc.).

This is again a discussion of terminology, which we have addressed above.

For the interface quality I commend the authors to have done high resolution TEM images, but I do not find them in the manuscript. In particular it would be great to have EELS data to check the interface (Fe or O terminated) of the YIG to understand the interface with the TI.

I agree in principle. Many things can be done along this line (XPS, x-STM, etc.). I should also point out that our main message is about the enhanced SSE from spin-momentum locked surface states in TIs tuned by varying Bi/Sb ratio.

I am a little skeptical about the data for the proximity. There are vastly different claims about the polarizability of for instance Pt (JHU vs. WMI, etc.). Actually I strongly believe that by doing XRMR and XMCD on the structures as a function, one could unambiguously determine the proximity effect. With contradicting data in the literature one cannot rely on literature but one must study the effect on the relevant samples. So as it stands now, I agree with the other referees, that the claims are too tenuous without further data.

This is actually a quantitative question. The magnetic proximity effect cannot be completely absent. In fact the mechanism we have explained for emf generation requires that there be some exchange coupling between YIG and the topological insulator surface states.

Furthermore I agree with Ref. #3 that the topological nature of the conversion cannot be

claimed from the data presented.

Please see Reviewer #3's new comments.

Furthermore the recent publication of the authors (Nano Lett., 2015, 15 (9), 5835) reveals many of the key materials features of the heterostructure used here.

Given that it is well known that the longitudinal spin Seebeck effect produces a spin current in YIG, the observed conversion into a charge signal is not that surprising given the already published results from the Nano Lett. paper and further literature that has shown the spin current to charge signal in TIs previously. The authors are probably right that the detailed study to disentangle the different contributions for the detection mechanism has not been presented and such a specialized study is of interested to people from the community.

As explained previously, this work establishes the mechanism of the emf generation by demonstrating its dependence on the position of the Fermi level within the topological insulator's surface states.

Finally as a minor point, concerning the claim: "(the DOI is incorrect but I know which work he/she refers to)": if from my side I enter the DOI I get to the paper that I referred to: <http://scitation.aip.org/content/aip/journal/apl/103/2/10.1063/1.4813315>

With the proximity induced magnetization issue not clarified in these samples and furthermore as it has been shown that TI materials can lead to apparently very high spin current to charge signal conversion efficiencies, I do not believe that Nature Materials is the natural outlet for this manuscript.

In our view the arguments he/she has presented in both rounds are unreasonable. If he/she does not believe the proximity effect in front of mounting evidence including ref. 25, it is ok, but his/her belief should not affect judging the SSE work since it was carried out well above the temperature where any controversy could arise. This work is not about generating spin current in YIG. Please do not think it is. It was also well beyond the scope of this work to demand us to demonstrate spin-orbit torque which by itself could be another separate paper.

A more specialized journal would possibly be more appropriate for a revised manuscript that focuses on the spin current detection mechanism and highlights the details of the different contributions.

Reviewer #3 (Remarks to the Author)

I have studied the remarks of the previous referees, together with the authors' reply, as well as the modifications the authors made to the manuscript.

I am satisfied with the reply of the authors to the remarks of referee 2. It is clear that at this stage not all questions can be answered. However, with the adjustments of the manuscript, I find the message of the authors is now sufficiently clear.

I am also satisfied with the reply of the authors to the remarks of referee 4. Indeed the meaning of "Spin Seebeck" effect is not well defined in literature. It could mean the generation of spin currents by a thermal gradient, or it could also refer to the actual (charge related voltages) which are finally generated and detected. The latter of course also includes the specific mechanism which converts the spin information into charge. In my opinion it is sufficiently clear that the authors address a new specific conversion of thermally generated spin currents in YIG to charge voltages. Also I agree with the authors that the message present manuscript is sufficiently new, as compared to earlier publications of them selves and others.

I can therefore recommend publication.

Reviewer #4 (Remarks to the Author)

The authors report large signals in a YIG/TI heterostructure where thermal spin currents are generated by a temperature gradient and these are then detected in the TI by an efficient spin to charge conversion. By carefully varying the TI materials composition they find that the amplitude depends on the position of the Fermi level with respect to the electronic states. Furthermore the authors exclude spurious Nernst effects as parasitic signal sources to unambiguously identify the origin of the signals.

Overall I believe this is the first demonstration of efficient detection of the thermal spin currents using TIs, which is thus an important finding that warrants publication. The key finding is the importance of the TI composition for the detection efficiency, while the spin Seebeck spin currents due to thermal gradients are well established by now.

I am surprised to see in the extended data figure 8 the claim "These results indicate the interface quality variation does not significantly affect the SSE magnitude". This is contrary to most findings in the community, where different interface treatments lead to different spin transmission and thus different signal amplitudes (see for instance overview of different interface treatments in APL 103, 22411 (2013)). Do the authors have an idea why in their case this does not seem to be the case? More recently interface effects (as well as bulk effects)

where suggested to have a strong influence on spin transmission across an interface (Nature Comms 7, 10452 (2016)). So given that the authors prepare their samples in 2 steps with a transfer in between, they must have established a reliable cleaning step before the deposition of the detection layer?

I also do not agree with the terminology "topological spin Seebeck effect", which is only mentioned once in the abstract. While the term spin Seebeck effect is not always coherently used in the literature, I think it should be made clear that there is nothing topological about the thermal magnon generation due to the temperature gradient but only the spin to charge conversion by the TI has a component that warrants the term "topological". So I would suggest to remove this phrase in the abstract and make this clearer in the text so that readers coming from different communities do not get confused.

I would just like to point out that there are experiments on the spin Seebeck effect that do not employ the inverse spin Hall effect (or a related spin-to-charge conversion mechanism) for the detection of the thermally generated spin currents. This includes a range of experiments from the Wees group (see A. Slachter et al., Nature Phys. 6, 879 (2010) and many related papers) where they detect the spin current in a non-local spin valve geometry by the spin-dependent chemical potential with respect to a second ferromagnet.

So one should add a paragraph in the description (for instance just before "In this letter..." - see my comment below), where the two steps, the generation of the spin current by the temperature gradient and the detection of the spin current by the spin to charge conversion are explicitly mentioned. Only together these two then result in the huge measured signals and the sophisticated part here is the latter.

Also my suggestion would be to slightly revise the title to:

Enhanced spin Seebeck effect signals due to spin-momentum locking in topological insulator surface states

This would be a true statement whatever one defines the spin Seebeck effect as.

Finally, I am not aware that Nature Communications publish "Letters", so the text should be revised accordingly.

So overall, the paper contains interesting new results that warrant publication. If my points are answered and the manuscript is made clearer with respect to the novel and sophisticated aspects, I would probably recommend publication in Nature Communications.

Reply to Reviewers' comments (original in blue and reply in black):

Reviewer #3 (Remarks to the Author):

I have studied the remarks of the previous referees, together with the authors' reply, as well as the modifications the authors made to the manuscript.

I am satisfied with the reply of the authors to the remarks of referee 2. It is clear that at this stage not all questions can be answered. However, with the adjustments of the manuscript, I find the message of the authors is now sufficiently clear.

I am also satisfied with the reply of the authors to the remarks of referee 4. Indeed the meaning of "Spin Seebeck" effect is not well defined in literature. It could mean the generation of spin currents by a thermal gradient, or it could also refer to the actual (charge related voltages) which are finally generated and detected. The latter of course also includes the specific mechanism which converts the spin information into charge. In my opinion it is sufficiently clear that the authors address a new specific conversion of thermally generated spin currents in YIG to charge voltages. Also I agree with the authors that the message present manuscript is sufficiently new, as compared to earlier publications of themselves and others. I can therefore recommend publication.

We thank the reviewer for the comments and recommendation.

To better answer 4th reviewer's long comments, we break into several parts and insert the replies to wherever is appropriate.

Reviewer #4 (Remarks to the Author):

The authors report large signals in a YIG/TI heterostructure where thermal spin currents are generated by a temperature gradient and these are then detected in the TI by an efficient spin to charge conversion. By carefully varying the TI materials composition they find that the amplitude depends on the position of the Fermi level with respect to the electronic states. Furthermore the authors exclude spurious Nernst effects as parasitic signal sources to unambiguously identify the origin of the signals.

Overall I believe this is the first demonstration of efficient detection of the thermal spin currents using TIs, which is thus an important finding that warrants publication. The key finding is the importance of the TI composition for the detection efficiency, while the spin Seebeck spin currents due to thermal gradients are well established by now.

We are glad that the main points of our work were appreciated by the reviewer. We thank the reviewer for his/her time and effort in reading our previous replies and explanations.

I am surprised to see in the extended data figure 8 the claim "These results indicate the interface quality variation does not significantly affect the SSE magnitude". This is contrary to most findings in the community, where different interface treatments lead to different spin transmission and thus different signal amplitudes (see for instance overview of different interface treatments in APL 103, 22411 (2013)). Do the authors have an idea why in their case this does not seem to be the case? More recently interface effects (as well as bulk effects) were suggested to have a strong influence on spin transmission across an interface (Nature Comms 7, 10452 (2016)). So given that the authors prepare their samples in 2 steps with a transfer in between, they must have established a reliable cleaning step before the deposition of the detection layer?

First of all, in the extended data figure 8, we showed two samples that were prepared with nominally the same procedures at two different times. The statement about the interface quality referred to those two samples whose interface quality probably does not differ as dramatically as in those samples that the reviewer was referring to, e.g. in APL 103, 22411 (2013). In that APL work, the samples were intentionally treated in widely different ways, e.g. by combinations of the "Piranha" etching method and Ar+ or O+/Ar+ plasma cleaning, etc. None of our samples had gone through different treatments like those. In addition, our YIG films are atomically flat films grown with PLD in an ultrahigh vacuum, rather than liquid phase epitaxy. The sample surface quality was examined with RHEED and AFM before and after every major step in the preparation as shown in the figures. Representative samples were chosen for high-resolution transmission electron microscopy (HRTEM). The HRTEM results already appeared in a paper which now cited as ref. 26. In contrast to the APL paper which does not show any AFM or HRTEM of their samples, we do know from our characterization data that the interface quality is good (shown in the figures in Extended Data and in the AIP Advances paper or ref. 26).

Second, as correctly pointed out by the reviewer, it has been established that interface quality greatly affects spin current transmission. In addition to the APL paper mentioned by the reviewer, two other papers by Saitoh's group (J. Phys. D 48, 164013 (2015) and APL 103, 092404 (2013)) contain very strong evidence. However, to generate poor interfaces, they had to resort to ion bombardment to create an amorphous layer (in J. Phys. D.) whose thickness was controlled by the acceleration voltage for ions. They concluded that the characteristic thickness of the amorphous layer is 2.3 nm. For 1 nm thick amorphous layer, the SSE magnitude is decreased by ~25%. In the APL paper (APL 103, 092404 (2013)), the amorphous layer is about 1 nm thick, but the spin pumping signal is dropped more dramatically (about one order of magnitude). The data from the same group indicate that the effect of the interface quality may be more serious on spin pumping than SSE! I must emphasize that in all three papers, the interface quality varies very dramatically via etching, ion bombardment, or annealing. On

contrary, we strove to keep the same processes and our HRTEM images on selected samples do not show any amorphous layer.

Third, most bilayer samples for SSE and spin pumping experiments (certainly in those samples for studying interfaces) contain micron-thick YIG films prepared by liquid phase epitaxy. Our YIG films are atomically flat and prepared with PLD in an ultrahigh vacuum which are free of large scale roughness or defects.

The procedures for growing TI are described in the first section of the Extended Data. Since the YIG films were freshly grown and not exposed to any solutions or contacted to solid substances, we performed annealing (600 C for 30 min) in the MBE chamber before TI deposition and monitored the quality with RHEED in-situ. The same procedures were adopted for all samples. We admit that this recipe is by no means an optimized one. Some degree of variation in interface quality is expected, which was attributed to the variation in the “Curie temperature” of the induced magnetization in TI at low temperatures (Extended data figure 4). The exchange coupling is more sensitive to the short-range defects or dirt; therefore, the “Curie temperature” variation is expected to be more sensitive to the variation in SSE.

We understand reviewer’s concern about this point. Instead of saying “interface quality variation does not significantly affect the SSE magnitude” which sounds like that we tried to negate the conclusions of previous studies, a better statement should be “These results indicate that the interface in these samples does not vary significantly to cause large variations in SSE magnitude”. Therefore, we specifically refer to our samples used in this study.

I also do not agree with the terminology "topological spin Seebeck effect", which is only mentioned once in the abstract. While the term spin Seebeck effect is not always coherently used in the literature, I think it should be made clear that there is nothing topological about the thermal magnon generation due to the temperature gradient but only the spin to charge conversion by the TI has a component that warrants the term "topological". So I would suggest to remove this phrase in the abstract and make this clearer in the text so that readers coming from different communities do not get confused.

Agreed. We removed the terminology along with the sentence in the abstract. We believe that the text is quite clear to readers about the special properties of the TI, i.e. the spin-momentum locking of the surface states. We do not find that any part in the text implies thermal magnon generation being topological.

I would just like to point out that there are experiments on the spin Seebeck effect that do not employ the inverse spin Hall effect (or a related spin-to-charge conversion mechanism) for the detection of the thermally generated spin currents. This includes a range of experiments from the Wees group (see A. Slachter et al., Nature Phys. 6, 879 (2010) and many related papers)

where they detect the spin current in a non-local spin valve geometry by the spin-dependent chemical potential with respect to a second ferromagnet.

We were aware of the paper and we agree with the reviewer. Unfortunately, the effect reported by Slachter et al., an elegant one, was not called "spin Seebeck effect", although it was a demonstration of thermally driven spin current. We are glad the reviewer acknowledged that there has been confusion in the terminology. We wish that the point had been communicated earlier.

So one should add a paragraph in the description (for instance just before "In this letter..." - see my comment below), where the two steps, the generation of the spin current by the temperature gradient and the detection of the spin current by the spin to charge conversion are explicitly mentioned. Only together these two then result in the huge measured signals and the sophisticated part here is the latter.

We added one sentence before "In this Letter" and changed "Letter" to "work". The added sentence reads "While pure spin current generation by heat is already established in bilayers consisting of a heavy metal and a ferromagnet, the detection of the spin current with topological surface states in TI has not been demonstrated."

Also my suggestion would be to slightly revise the title to:

Enhanced spin Seebeck effect signals due to spin-momentum locking in topological insulator surface states. This would be a true statement whatever one defines the spin Seebeck effect as.

We changed the title as suggested.

Finally, I am not aware that Nature Communications publish "Letters", so the text should be revised accordingly.

It has been changed.

So overall, the paper contains interesting new results that warrant publication. If my points are answered and the manuscript is made clearer with respect to the novel and sophisticated aspects, I would probably recommend publication in Nature Communications.